# Validity and Reliability of the Caregiving Difficulty Scale in Mothers of Children with Cerebral Palsy

**DOI:** 10.3390/ijerph18115689

**Published:** 2021-05-26

**Authors:** Eun-Young Park

**Affiliations:** Department of Secondary Special Education, College of Education, Jeonju University, Jeonju 55069, Korea; eunyoung@jj.ac.kr; Tel.: +82-63-220-3186

**Keywords:** Caregiving Difficulty Scale, mothers, children with cerebral palsy, validity, reliability

## Abstract

This study was conducted to determine the construct validity and reliability of the Caregiving Difficulty Scale, a tool developed to measure difficulties experienced by parents of children with cerebral palsy. To this end, a survey was conducted with 215 mothers of children with cerebral palsy, and the resultant data were analyzed. A confirmatory factor analysis was performed to verify the construct validity of this scale, and the intra-item fit value was calculated for reliability analysis. Validity analysis confirmed that a bi-factor model comprising four sub-factors, Concern for the Child, Impact on Self, Support for Caregiving, and Social and Economic Strain, was suitable for the Caregiving Difficulty Scale. In addition, the reliability analysis results showed that the reliability coefficients of three of these areas, excluding Social and Economic Strain, and the reliability of the entire scale were acceptable. Therefore, the Caregiving Difficulty Scale is an appropriate tool to measure the burden of caregiving for children with cerebral palsy, and the findings emphasize the need to improve its reliability by comparing sub-factors’ reliability.

## 1. Introduction

Caring for a child is one of the normal parental roles; however, excessive demands related to caring for a child with disabilities can lead to increased burden or stress among caregivers [1,2]. Caring for children with chronic conditions is often associated with negative health outcomes, such as depression, stress, anxiety, and low self-efficacy, for caregivers [1,3,4,5,6,7]. In addition, the caregiving burden can instigate psychological changes including depression, insomnia, and loss of motivation [8,9]. This was demonstrated in a study by Clyburn and others [10], who found that the caregiving burden stimulated psychological changes including depression and related symptoms. Caregiving burden has also been reported as one of the major causes of a reduction in subjective psychological well-being and life satisfaction in caregivers [11,12]. Furthermore, a Canadian population-based study showed that caregivers of children with health problems, compared to caregivers of healthy children, were twice as likely to report chronic conditions, activity limitations, and elevated depressive symptoms, and were also more likely to report poorer general health [13]. Studies have also reported an increase in the mortality rate of caregivers [14].

Cerebral palsy (CP), a common neuromotor disorder, is one of the most common childhood chronic diseases to cause long-term functional limitations [15,16]. The prevalence of CP has been estimated to be 2.08 per 1000 live births [17]. It is a non-progressive abnormality of fetal or infancy brain development and causes permanent impairment of movement and posture development due to brain lesions. Children with CP show the most complex disabilities and experience limitations in activities and participation [18,19,20]. Chronic and recurrent pain is one of the problems seen in children with CP, and it plays an important role in the quality of life of such children and their parents [21,22]. The long-term effects of these disabilities and accompanying problems in children with CP are more pronounced. In addition to movement disorders, CP has secondary concomitant disorders, such as sensory, perception, cognition, communication, and behavioral disorders, as well as epilepsy and secondary musculoskeletal problems [23]. As is evident, problems in children with CP are complex, long-term, and often severe [24,25]. Therefore, most children with CP need lifelong support in their daily routine activities [19,26,27]. However, long-term care for children with CP negatively affects the daily lives of children and their parents [6,28,29]. To illustrate this, studies on mothers of children with CP have reported that the degree of depression, anxiety, and stress was generally higher than that of mothers without children with CP [29,30,31].

Caregiving burden was identified as one of the stress-inducing factors that impact parents’ well-being [11,31]. Caregiving burden is a physical, psychological, emotional, social, and economic problem experienced by a family caring for a patient with a disability and is useful to measure the subjective stress of a caregiver. Among the variables related to disability characteristics, the type and degree of the disability have been reported as factors influencing parenting burden [28,32]. CP, which has complex disability characteristics, is one of the representative diseases about which caregivers complain, in that it causes difficulty in raising their children. Thus, assessing caregiver health outcomes is essential, as they are a valuable resource in the rehabilitation of children with long-term disabilities [6,7]. Since caring for a family member with a disability is known to affect caregiver health, it is important to regularly assess factors that cause burden/stress for the caregiver [6]. This can only be achieved by measuring psychometrically sound outcomes [33]. Therefore, to provide adequate support and services to caregivers, it is necessary to manage the care burden imposed on the caregiver, which can only be achieved if there is an appropriate measure to assess the caregiver’s burden [2,7,34].

The burden of support for parents and families raising children with CP has been a major topic of research. Marrón et al. [35] reported a significant relationship between self-efficacy, degree of disability, and depression in their study of the factors related to the burden of care for 62 caregivers, and these variables accounted for 40.9% of the burden of care. However, in previous studies of the burden of caring for children with CP, there are problems regarding the tools used to measure the burden of support. Marrón et al. [35] used the Zarit Burden Interview, a tool developed to measure the difficulty in caring for older patients such as those with dementia [36]. In a recent study [37] that investigated the burden of care for parents of children with CP, a self-developed tool comprising four questions and an item on time spent caring was used. However, that study did not confirm the validity of the measure. While burden can be assessed qualitatively, it is beneficial to have an easy-to-use, proven tool that can quantify the degree of burden on a caregiver and can provide an objective means of follow up. The Caregiver Difficulty Scale (CDS) was developed to measure difficulties in caring for children with CP [2]. Thereafter, the CDS was translated into Persian and its psychometric characteristics were confirmed [16]. Considering the lack of tools to measure the burden of caring for children with CP, studies to adapt developed tools and confirm their validity are essential before their application in practice. Thus, the purpose of this study was to translate the CDS into Korean and to verify its construct validity and reliability in measuring the burden associated with caring for children with CP.

## 2. Materials and Methods

### 2.1. Participants

The participants of this study were conveniently sampled from among 215 mothers of children with CP who were using community welfare centers or receiving treatment at rehabilitation hospitals or rehabilitation centers. Participants were sampled from all the regions of South Korea except for Jeju Island. The purpose and procedure of the study were first explained to the children, and their mothers’ consent was obtained. This study was part of a three-year project, and CDS data were collected the previous year (IRB−1041042−2013−1). Informed consent was obtained from all subjects involved in the study. Most mothers were aged 40–49 years (59.0%). In terms of education, college graduates accounted for more than half (76.7%), and 69.8% of mothers did not have a job.

Further, more than half of the children (62.8%) were boys, and their average age was 8.5 years (SD = 3.6). The types of CP included spastic (76.7%), dyskinetic (13.5%), and ataxic (9.8%). Using the Gross Motor Function Classification System (GMFCS) distribution, 11.6% were classified as Level 1, 9.8% as Level 2, 7.4% as Level 3, 16.7% as Level 4, and 54.5% as Level 5. A full overview of the characteristics of the participants is displayed in Table 1 below.

### 2.2. Measure

#### 2.2.1. Gross Motor Function Classification System (GMFCS)

The GMFCS is used to classify the gross motor function of children with CP. The big action function classification system comprises five levels. Similarly, the GMFCS system is divided into five levels, ranging from Level 1 to 5 as follows: Level 1 refers to being able to walk without any restrictions; at Level 2, an individual can walk with restrictions; in Level 3, individuals can walk with a cane, a pair of crutches, or walking aids without body support. Level 4 is classified as self-moving by means of transportation, and Level 5 refers to serious limitations in mobility. The inter-rater reliability of the GMFCS was reported to be 0.84 in a previous study [38].

#### 2.2.2. Caregiving Difficulty Scale

The CDS contains 25 items and is divided into four sub-factors, namely, Concern for the Child (8 items), Impact on Self (7 items), Support for Caregiving (5 items), and Social and Economic Strain (5 items). Each item is scored on a 5-point Likert scale (0–4) representing the frequency/range of each care experience perceived by the caregiver; the final total score ranges from 0 to 100. The higher the score, the heavier the burden on the caregiver’s life [2]. To determine the validity and reliability of the CDS to measure mothers’ burden of care for children with CP, an adaptation of the CDS (an evaluation tool) was implemented following the standard adaptation process of adaptation, review, reversal, and review [39,40]. Two special education majors modified the CDS and reviewed whether there were any questions that did not fit the context of Korea. No questions were deemed unsuitable for the Korean context. After discussing and combining the contents of the proposed version, the reverse version (the version translated back into English) was implemented. The reverse draft was prepared by a native English speaker who did not major in a field related to education. The results of the reversal were compared with the original tool and reviewed by commissioning three related experts (in special education and physical therapy).

### 2.3. Statistical Analysis

Normality of data was examined using skewness and kurtosis values. Cases in which the absolute value of skewness was above 3 and the kurtosis exceeded 8 or 10 were considered to be extreme [41]. The skewness and kurtosis of all items of the CDS were below the absolute value of 2. The maximum absolute value was 1.16 for skewness and 1.24 for kurtosis.

In this study, to verify the factor structure for the CDS translated into Korean for mothers of children with CP, a confirmatory factor analysis was performed on the four-factor structure identified in a previous study [16]. For this, data analyses were performed using the statistical program AMOS 25.0 (IBM Inc., Chicago, IL, USA). The comparative fit index (CFI) and the root mean square error of approximation index (RMSEA) were used as indices to determine the fit of the model [42]. CFI are values that show how good the model to be evaluated is compared to the null model, and if it is 0.9 or more, it is considered to be a model with good fit. The RMSEA value is an absolute goodness-of-fit index that shows how well the set model fits the data without comparing it with other models. If it is less than 1.0, it is judged to be a suitable model [43]. The reliability of the CDS was assessed by calculating the internal consistency of Cronbach’s α.

## 3. Results

### 3.1. CFA of the CDS

Table 2 presents the fit indices for the CDS. The fit indices provide evidence for the bi-factor model (4 factors within one factor) of mothers of children with CP. Furthermore, the CFI was above 0.90 and RMSEA was below 0.10.

Regression weights for the CDS-based bi-factor model are presented in Table 3. As shown in Table 3, regression weights for 23 items were significant at *p* = 0.01, and two items were significant at *p* = 0.05.

### 3.2. CDS Reliability

As shown in Table 4, the overall scale reliability was good (internal consistency using Cronbach’s α = 0.892; 95% confidence interval: 0.870–0.912). The reliability of Concern for the Child, Impact on Self, and Support for Caregiving was good (internal consistency using Cronbach’s α = 0.823, 0.849 and 0.767, respectively). The reliability of Social and Economic Strain was low (internal consistency using Cronbach’s α = 0.687).

## 4. Discussion

Providing care for children with CP has been shown to increase caregiving burden and lead to numerous health issues; however, little research has determined the validity and reliability of measures to assess this in Korea. Therefore, this study conducted a survey of mothers of children with CP to determine whether the CDS translated into Korean was a valid tool for measuring caring difficulties experienced by mothers of children with CP. The results confirmed the factor structure of this measure through CFA, and the reliability of the sub-factors was verified.

When determining the construct validity, the goodness-of-fit of the CFI was low, and the RMSEA was good. In the context of the CFA, it is recommended to use RMSEA, which is not sensitive to sample size and the simplicity of the model is preferred as the criterion for determining the goodness of fit of the model [42]. Therefore, in this study, in addition to a low CFI index, a bi-factor model including four sub-factors was accepted based on the confidence intervals of the RMSEA goodness-of-fit index. In addition, in a study that investigated the validity of the CDS after adapting it into Persian, the goodness-of-fit indices of CFI, TLI, and RMSEA were all satisfactory [16]. Cutoffs for a fit index can be misleading and subject to misuse [44]. Although the chi-squared test is widely used to analyze model fit, it was not used as a fit statistic in the present study because it is sensitive to sample size. Furthermore, it is not optimal to strive for single-test accept/reject decisions because the nature of such tests is very different from conventional hypothesis tests such as the t-test. Hence, it is important to use other goodness-of-fit measures to determine global model fit and attend to diagnostics for the sources of model misfit. RMSEA and CFI are commonly used to measure fit [45].

In the analysis of the internal reliability of all the items and subdomains of the CDS, the reliability coefficient (Cronbach’s α) for all items was 0.892, and the reliability coefficients for each subdomain ranged from 0.687 to 0.849. Since reliability can be considered acceptable when the Cronbach’s α is 0.7 or more, and excellent when it is 0.9 or more [46], all items of the CDS were deemed reliable. However, the subdomain of Social and Economic Strain was statistically very suitable. In this subdomain, excluding items 23 and 24 improved the reliability coefficient of the CDS to an acceptable value of 0.711. Therefore, when using the scores of the subdomains of the CDS in this study, it would be appropriate to use the total score as the question of the reliability of the Social and Economic Strain subdomain score may be raised. However, the low reliability of this subdomain suggests that additional psychometric studies are needed to compare the reliability of the subdomains.

This study has some limitations. A classic verification method of scale development is performing a factor analysis. However, even in the case of the items derived through the CFA, attempts to verify scales by other statistical methods are required to accurately evaluate the suitability and difficulty of the items. For example, by applying the Rasch model, it is possible to select duplicate items or items that are less suitable for each subdomain from among the developed items. Furthermore, this model can determine whether the items are questioning the concepts to be measured, from a high to low level. This is a research method that increases the degree of completion to the degree that has been verified by factor analysis, and increases the need to remove or correct items by applying a more stringent standard. Therefore, in the future, a study to confirm the validity of CDS through Rash analysis is necessary to further verify the suitability of the CDS in measuring caregiver burden. Another limitation was related to the sex of participants. The sex ratio of selected mental disorders, such as major depressive disorders and anxiety disorders, is higher in women than in men [47]. Since sex differences may exist in the degree of caregiving burden, this study, which included only mothers, needs to be expanded in the future to include fathers. Lastly, the results of a comparison with another gold standard scale are needed. Since the purpose of this study was to determine the construct validity and reliability of the CDS, comparison results with other measures were not presented, but it is necessary to confirm the concurrent validity, another aspect of the validity.

## 5. Conclusions

The purpose of this study was to adapt the CDS to the Korean context, to measure caregiver difficulty in raising children with CP, and to confirm the reliability and validity thereof. To this end, the factor structure and reliability of CDS were investigated in mothers of children with CP, and a bi-factor model comprising four sub-factors of Concern for the Child, Impact on Self, Support for Caregiving, and Social and Economic Strain was found to be an appropriate structure for measuring the difficulty of caring for a child with CP. The reliability coefficients of three areas, excluding Social and Economic Strain, and the reliability of the entire scale were acceptable. Therefore, future research should focus on comparing the reliability of the subdomains of the CDS to verify its overall reliability. These findings provide important insights into the validity and reliability of the CDS for measuring caregiver burden, which can be used in future developments of this scale as well as in its successful implementation to determine specific areas of caregiver burden.

## Figures and Tables

**Table 1 ijerph-18-05689-t001:** General characteristics of participants.

Category	Sub-Category		*n*	%
Children	Sex	Boy	135	62.8
		Girl	80	37.2
	Type	Spastic	165	76.7
		Dyskinetic	59	13.5
		Ataxic	21	9.8
	GMFCS	Level 1	25	11.6
		Level 2	21	9.8
		Level 3	16	7.4
		Level 4	36	16.7
		Level 5	117	54.4
Mothers	Age	30~39	78	36.3
		40~49	127	59.0
		50≤	10	4.7
	Education Level	Graduate college	165	76.7
		Graduate high school	44	20.5
		Graduate middle school	2	0.9
		Missing value	4	1.9
	Employment	Yes	61	28.4
		No	150	69.8
		Missing value	4	1.9

**Table 2 ijerph-18-05689-t002:** Model Fit Index of the CDS.

df	χ^2^	CFI	RMSEA (LO 90~HI 90)
270	528.076	0.877	0.067 (0.0586~0.075)

CFI = Comparative Fit Index; RMSEA = root mean square error of approximation.

**Table 3 ijerph-18-05689-t003:** Regression Weights of Two-Factor Model.

Parameter Estimated Value	B	SE	CR
CDS ← Concern for the Child (CC)	1		
CDS ← Impact on Self (IS)	5.19	1.86	2.790 **
CDS ← Support for Caregiving (SC)	1.694	0.700	2.420 *
CDS ← Social and Economic Strain (SES)	5.43	1.978	2.745 **
CC ← 1. Does your child fall ill from time to time?	1		
CC ← 2. Are you satisfied about the improvement in your child’s condition after receiving treatment/therapy?	1.542	0.629	2.450 *
CC ← 3. Do you fear what your child’s future might be?	3.249	1.124	2.891 **
CC ← 4. Do you worry about your child’s present state?	2.91	1.012	2.876 **
CC ←5. Do you worry that your child cannot function like other children (e.g., going to school, playing)?	3.817	1.317	2.898 **
CC ← 6. Do you feel sad that your child cannot do anything by himself/herself?	4.02	1.392	2.888 **
CC ← 7. Do you worry that your child gets insulted and/or ridiculed by others?	4.239	1.453	2.918 **
CC ← 8. Do you fear that your child will have accidents as a result of his/her disability?	4.377	1.501	2.915 **
IS ← 9. Does caring for the child make you feel tired and exhausted?	1		
IS ← 10. Does the child’s condition prevent you from being relaxed?	1.121	0.093	12.102 **
IS ← 11. Do you have enough time to look after your own health?	0.564	0.085	6.612 **
IS ← 12. Do you have enough time for your basic daily needs such as having meals, sleeping, bathing, etc.?	0.68	0.091	7.455 **
IS ← 13. Do you feel that you will never have enough time to get everything done?	0.899	0.090	9.994 **
IS ← 14. Do you think that your health has been affected because of your child’s condition?	0.982	0.089	10.979 **
IS ← 15. Does the child’s condition prevent you from attending to the needs of other family members?	0.927	0.089	10.438 **
SC ← 16. Does your spouse help you with the care of this child?	1		
SC ← 17. Does your spouse support you in other family responsibilities?	1.06	0.181	5.865 **
SC ← 18. Are you able to discuss your child’s problems with other family members?	1.744	0.245	7.109 **
SC ← 19. Are the other family members well aware about the child’s condition?	1.389	0.200	6.941 **
SC ← 20. Do your relatives/neighbors help you with caring for the child?	1.173	0.214	5.490 **
SES ← 21. Do you have to restrict your social visits and relationships due to the child’s illness?	1		
SES ← 22. Do you have to face embarrassing situations when you are traveling with the child?	0.995	0.121	8.200 **
SES ← 23. Is there an increase in your family expenses due to the child’s condition?	0.899	0.120	7.465 **
SES ← 24. Is your income adequate to provide the necessities for the child?	0.531	0.099	5.338 **
SES ← 25. Do you worry that you are unable to provide special facilities needed by your child?	0.419	0.106	3.968 **

Note. B = Non-standardized coefficient; SE = Standard Error; CR = Critical Ratio; * *p* < 0.05; ** *p* < 0.01.

**Table 4 ijerph-18-05689-t004:** Internal Consistency of the CDS.

Factor	Cronbach α	95% CI
Concern for the Child	0.823	0.784–0.856
Impact on Self	0.849	0.816–0.878
Support for Caregiving	0.767	0.714–0.813
Social and Economic Strain	0.687	0.615–0.748
Total 25 items	0.892	0.870–0.912

Note. CI = confidence interval.

## Data Availability

The data presented in this study are available on request from the corresponding author.

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
