# Peer review of "Validity and Reliability of the Caregiving Difficulty Scale in Mothers of Children with Cerebral Palsy"

_ijerph, 2021, doi:10.3390/ijerph18115689_

Round 1

Reviewer 1 Report

Introduction :

This landmark study showing increase mortality for these caregivers should be mentioned here :

Early mortality and primary causes of death in mothers of children with intellectual disability or autism spectrum disorder: a retrospective cohort study. Fairthorne J, Hammond G, Bourke J, Jacoby P, Leonard H.PLoS One. 2014 Dec 23;9(12):e113430. doi: 10.1371/journal.pone.0113430. eCollection 2014.

This was demonstrated by Cliburn, Stones   ( 10) …. Change to : This was demonstrated by Cliburn and others ( 10)

“Cerebral Palsy (CP), a common neuromotor disorder, is one of the most common childhood chronic diseases that causes long-term functional limitations [14-16]. “  A number of prevalence ( 2.5-1000 ) with an appropriate reference should be given at this point to give to the reader a better dimension of the phenomenon. : MaÅ‚gorzata Sadowska,1 Beata Sarecka-Hujar,2 and Ilona Kopyta Neuropsychiatr Dis Treat. 2020; 16: 1505–1518. Cerebral Palsy: Current Opinions on Definition, Epidemiology, Risk Factors, Classification and Treatment Options

The issue of chronic and recurrent pain plays a major role on the quality of life of these patients and parents, this should be mentioned as referenced as well with two references : Pain in cognitively impaired children: a focus for general pediatricians. Massaro M, Pastore S, Ventura A, Barbi E. Eur J Pediatr. 2013 Jan;172(1):9-14. doi: 10.1007/s00431-012-1720-x. Epub 2012 Mar 20.

Developing a Sense of Knowing and Acquiring the Skills to Manage Pain in Children with Profound Cognitive Impairments: Mothers' Perspectives. Carter B, Arnott J, Simons J, Bray L.Pain Res Manag. 2017;2017:2514920. doi: 10.1155/2017/2514920. Epub 2017 Mar 26.

“ In a recent study [34] that investigated the burden of parenting for children with CP, a self-developed re- search tool consisting of four questions and time spent caring was used; this was not a valid tool “   Please explain why this is not a validated tool and justify the statement .

Material and methods . Partecipants. How were mother selected and  contacted ? Did some of them refuse to participate ?

Author Response

Dear reviewer

Thank you very much for giving me the opportunity to revise our manuscript. I am grateful to you for insightful suggestions and comments. I have carefully reviewed the comments and incorporated them to strengthen our manuscript.

In the revised manuscript, I have highlighted in blue where we have made changes; please, note that I removed track changes that show all my edits.

The details of the changes are provided below this letter.

Thank you again for your time and efforts to review my manuscript and to provide insightful suggestions and edits.

Comment #1: This landmark study showing increase mortality for these caregivers should be mentioned here:

Early mortality and primary causes of death in mothers of children with intellectual disability or autism spectrum disorder: a retrospective cohort study. Fairthorne J, Hammond G, Bourke J, Jacoby P, Leonard H.PLoS One. 2014 Dec 23;9(12):e113430. doi: 10.1371/journal.pone.0113430. eCollection 2014.

 Response #1: Thank you for your comment. It has been mentioned and inserted citation.

Studies have also reported an increase in the mortality rate of caregivers [14].

Comment #2: This was demonstrated by Cliburn, Stones   ( 10) …. Change to : This was demonstrated by Cliburn and others ( 10)

Response #2: It have been revised according to comment.

This was demonstrated in a study by Clyburn and others [10],

Comment #3: “Cerebral Palsy (CP), a common neuromotor disorder, is one of the most common childhood chronic diseases that causes long-term functional limitations [14-16]. “  A number of prevalence ( 2.5-1000 ) with an appropriate reference should be given at this point to give to the reader a better dimension of the phenomenon. : MaÅ‚gorzata Sadowska,1 Beata Sarecka-Hujar,2 and Ilona Kopyta Neuropsychiatr Dis Treat. 2020; 16: 1505–1518. Cerebral Palsy: Current Opinions on Definition, Epidemiology, Risk Factors, Classification and Treatment Options

Response #3: It have been revised according to comment.

The prevalence of CP has been estimated to be 2.08 per 1000 live births [17].

Comment #4: The issue of chronic and recurrent pain plays a major role on the quality of life of these patients and parents, this should be mentioned as referenced as well with two references : Pain in cognitively impaired children: a focus for general pediatricians. Massaro M, Pastore S, Ventura A, Barbi E. Eur J Pediatr. 2013 Jan;172(1):9-14. doi: 10.1007/s00431-012-1720-x. Epub 2012 Mar 20.

 Developing a Sense of Knowing and Acquiring the Skills to Manage Pain in Children with Profound Cognitive Impairments: Mothers' Perspectives. Carter B, Arnott J, Simons J, Bray L.Pain Res Manag. 2017;2017:2514920. doi: 10.1155/2017/2514920. Epub 2017 Mar 26.

 Response #4: It have been revised according to comment.

Chronic and recurrent pain is one of the problems seen in children with CP, and it plays an important role in the quality of life of such children and their parents [21, 22].

Comment #5: “In a recent study [34] that investigated the burden of parenting for children with CP, a self-developed re- search tool consisting of four questions and time spent caring was used; this was not a valid tool “  Please explain why this is not a validated tool and justify the statement .

Response #6: Reason have been described.

In a recent study [37] that investigated the burden of care for parents of children with CP, a self-developed tool comprising four questions and an item on time spent in caring was used. However, that study did not confirm the validity of the measure.

Comment #6: Material and methods . Partecipants. How were mother selected and contacted ? Did some of them refuse to participate?

Response #7: Sentence has been revised. For the purpose of recruiting research subjects, the purpose and procedure of the study were first explained to the target children and mothers, and consent was sought. Survey was conducted only when agreed.

The purpose and procedure of the study were first explained to the children, and their mothers’ consent was obtained.

Reviewer 2 Report

  • Authors must remove space in references.
  • Authors must explain with more detail where the sample was collected (city, region).
  • Authors must explain why the sample is composed only by women. Women many times have reported higher level of anxiety than men. Are those gender differences influence the results of this study?
  • Authors could explain why they do not use another scale as a golden  standard to prove the utility of CDS.

Author Response

Dear reviewer

Thank you very much for giving me the opportunity to revise our manuscript. I am grateful to you for insightful suggestions and comments. I have carefully reviewed the comments and incorporated them to strengthen our manuscript.

In the revised manuscript, I have highlighted in blue where we have made changes; please, note that I removed track changes that show all my edits.

The details of the changes are provided below this letter.

Thank you again for your time and efforts to review my manuscript and to provide insightful suggestions and edits.

Comment #1: Authors must remove space in references.

Response #1: It have been removed.

Comment #2: Authors must explain with more detail where the sample was collected (city, region).

Response #2: It has been mentioned.

Participants were sampled from all regions in South Korea except for Jeju Island.

Comment #3: Authors must explain why the sample is composed only by women. Women many times have reported higher level of anxiety than men. Are those gender differences influence the results of this study?

Response #3: The limitation and recommendation for further study regard to sex differences have been added in discussion. 

Another limitation was related to the sex of participants. The sex ratio of selected mental disorders, such as major depressive disorder and anxiety disorder, is higher in women than in men [46]. Since sex differences may exist in the degree of caregiving burden, this study, which included only mothers, needs to be expanded in the future to include fathers.

Comment #4: Authors could explain why they do not use another scale as a golden standard to prove the utility of CDS.

Response #3: The purpose of the study was revised to limit it to confirmation of construct validity. The limitation and recommendation for further study regard to concurrent validity.

Please, see throughout manuscript including title.

Lastly, the results of a comparison with another gold standard scale are needed. Since the purpose of this study was to determine the construct validity and reliability of the CDS, comparison results with other measures were not presented, but it is necessary to confirm the concurrent validity, another aspect of the validity.